# Symbolic Planning Using LLM Agents: A Cut-Based Reprompting Approach

## Abstract

Large Language Models (LLMs) are increasingly used for symbolic planning, yet their stochastic outputs can lead to invalid results. We study propose–validate tasks where candidate checking is cheap – e.g., a single traversal of a finite state machine (FSM), or a short simulator rollout – but finding a valid plan is non-trivial. We pair an LLM decision maker with a symbolic validator via cut-based reprompting: the model proposes a plan; the validator checks it and returns structured cuts that forbid the observed failure patterns; the model is reprompted with the accumulated cuts. We formalize LLM planning as sampling from a context-conditioned stochastic policy and show that symbolic cuts monotonically reduce policy entropy and guarantee finite-step convergence under mild assumptions inherent to FSM modeling. We empirically assess cut-based reprompting on FSM traversal problems and MiniGrid, observing higher success probability and faster convergence (fewer reprompts) than vanilla reprompting for GPT-4o-mini and LLaMA3-8. Beyond accuracy, this pattern offers zero-shot flexibility relative to bespoke search: evolving exceptions and task-specific rules can be injected as textual input without modifying a solver. Overall, cut-based reprompting provides a general, plan-level mechanism for making LLM planners more fault-tolerant, interpretable, and controllable.

## 1 Introduction

Large Language Models (LLMs) excel at compositional reasoning and planning but remain inherently stochastic: identical queries can yield divergent outputs, many violating hard constraints. This unreliability limits their deployment in structured domains such as graph traversal, symbolic reasoning, and sequential decision-making, where correctness is essential. Existing prompting strategies—chain-of-thought, tree-of-thought, self-reflection—offer partial improvements but operate at the token level and rely on self-critique, which is unreliable for smaller models.

Classical planners excel when specifications are fixed and fully encoded, but real-world task definitions accumulate exceptions ("never pass through locked nodes except after key-pickup," "avoid revisiting hazard states unless sensor X is true," "respect temporary maintenance blocks"). Encoding and maintaining these bespoke rules inside a search stack can be brittle and costly. Our approach keeps the planner generic and pushes exceptions into textual, zero-shot constraints (cuts) that the validator can generate automatically, preserving flexibility while retaining symbolic guarantees.

We take a different view by modeling LLMs as *context-conditioned stochastic policies*. The prompt defines a distribution over candidate plans, and decoding corresponds to sampling from it. A persistent challenge is that non-trivial probability mass is assigned to invalid solutions, leading to repeated errors. To address this, we introduce *cut-based reprompting*, a lightweight context-engineering method inspired by cutting-plane techniques. After each invalid attempt, an external validator extracts symbolic constraints (*cuts*) that forbid incorrect transitions and appends them to the prompt. Each iteration shrinks the policy's support, reduces entropy, and steers the model toward valid completions. Stronger cuts generalize feedback by pruning all invalid successors of a state, accelerating convergence.

We develop a theoretical framework showing that symbolic cuts monotonically reduce support size (Hartley entropy), guarantee convergence within a bounded number of iterations, and sharpen the posterior distribution over feasible plans. Empirically, we evaluate across synthetic DAGs, grids,

and task graphs (10–100 nodes). With GPT-4o-mini and LLaMA-3-8B, cut-based strategies consistently improve solve rates over vanilla prompting and naive reprompting. Entropy analysis confirms monotone convergence, while smaller models reveal a bias–capacity tradeoff in exploiting symbolic feedback.

**FSMs as a General Abstraction.** Although our experiments focus on FSM traversal and MiniGrid Chevalier-Boisvert et al. (2023), the underlying abstraction is deliberately broad. A wide range of planning problems can be cast as state machines, where nodes represent world configurations and edges encode feasible actions. This includes domains such as robotics task execution, workflow scheduling, web navigation, software verification, and program synthesis. Thus, our formulation is not restricted to synthetic graphs: FSM traversal serves as a canonical testbed that isolates the reasoning dynamics while retaining relevance to real-world sequential decision-making.

## 2 RELATED WORK

**LLMs as Planners.** Prior work has applied LLMs to structured planning tasks with strict preconditions. Systems such as PROGPROMPT Singh et al. (2023), LLM-GENPLAN Silver et al. (2024), and ADAPLANNER Sun et al. (2023) synthesize plans from declarative specifications, often leveraging domain knowledge or explicit transition models. Other works ground outputs symbolically (G-PLANET Lin et al. (2023), PLAG Lin et al. (2024)) or emphasize symbolic execution (CHAIN-OF-SYMBOLS Hu et al. (2024), PPNL Aghzal et al. (2023)), aligning with our FSM traversal setting but without feedback-driven correction.

**Constrained and Structured Decoding.** Grammar- and schema-constrained decoding restricts outputs to CFGs or schema-valid forms Geng et al. (2023); Park et al. (2024). These ensure local syntactic validity but require task-specific grammars and cannot guarantee global path feasibility. Cut-based reprompting instead enforces *plan-level* validity by pruning infeasible transitions, making it complementary to constrained decoding.

**Feedback and Repair Loops.** Iterative prompting methods such as REFLEXION Shinn et al. (2023), SELF-REFINE Madaan et al. (2023), and INSTRUCT-OF-REFLECTION Liu et al. (2025) rely on self-critique, which is often unreliable for smaller models. Verifier-guided and repair-based methods Li et al. (2023); Yao et al. (2023) use external checks but act locally at the token or edit level. Our approach differs by validating entire plans and injecting symbolic constraints, yielding monotone entropy reduction.

**LLMs as Heuristics for Search.** Recent work Valmeekam et al. (2023); Wang et al. (2024) uses LLMs as heuristics to guide classical planners. While effective, these approaches rely on external solvers. In contrast, cut-based reprompting is solver-free, lightweight, and model-agnostic, yet still provides convergence guarantees.

**Policy Shaping and Entropy Control.** In reinforcement learning, policy shaping and entropy regularization balance exploration and convergence Abdolmaleki et al. (2018). We connect this perspective to LLM planning: symbolic cuts act as entropy filters that monotonically reduce support size and sharpen distributions toward feasible solutions.

**Why not just decode with constraints?** A natural question is whether constrained decoding alone suffices. While it enforces syntax, it cannot capture task-specific feasibility constraints (e.g., graph reachability, avoiding dead ends), and is brittle when constraints depend on dynamic context. Cut-based reprompting, by contrast, is prompt-based, model-agnostic, and generalizes across domains by pruning infeasible plans iteratively.

**Our Contribution.** We differ from prior work in three respects: (1) we introduce *cut-based reprompting*, a lightweight feedback loop that enforces symbolic feasibility at the plan level; (2) we provide a theoretical framework proving monotone entropy reduction and bounded convergence; and (3) we demonstrate applicability across FSM traversal and MiniGrid, highlighting both performance gains and diagnostic insights into LLM reasoning limits.

# 3  THEORETICAL FRAMEWORK

We analyze cut-based reprompting through the lens of entropy reduction in stochastic policies. At iteration $t$, an LLM defines a probability distribution $\pi_\theta(P \mid C_t)$ over candidate paths $P$, conditioned on context $C_t$. Let $\mathcal{S}_t \subseteq \mathcal{P}$ denote the feasible support after $t$ iterations, where $\mathcal{P}$ is the set of all paths from $s_0$ to $s_G$.

**Definition 1 (Cut).** A *cut* $c$ is a symbolic constraint that removes a subset of paths from $\mathcal{S}_t$. Formally,
$$\mathcal{S}_{t+1} = \{P \in \mathcal{S}_t : P \notin c\},$$
where $P \in c$ means that path $P$ violates the constraint. For example, forbidding edge $(s_i, s_j)$ removes all paths containing that edge. Weak cuts forbid only specific invalid transitions, while strong cuts enumerate valid successors of a node and forbid all other outgoing edges.

**Assumptions.** Our analysis relies on the following: (A1) **Sound validator:** inherent to FSM modeling, feasibility of a path can always be checked by traversing the transition structure. (A2) **Stable prompts:** cuts are faithfully incorporated into the context. (A3) **Consistency:** cuts do not eliminate all valid solutions (or a recovery mechanism exists).

**Remark.** The critical modeling choice is to cast planning problems as FSMs or state-transition systems. Under this abstraction, the existence of a validator is not an external assumption but an inherent property: path validity can always be determined by traversing the FSM. The key assumption is therefore that the task of interest admits a tractable FSM representation.

**Why Hartley entropy?** A natural measure of uncertainty is Shannon entropy. However, removing low-probability paths can paradoxically *increase* Shannon entropy after renormalization, since the remaining distribution may become more uniform (Appendix A gives a counterexample). In contrast, Hartley entropy
$$H_0(\mathcal{S}_t) = \log |\mathcal{S}_t|$$
depends only on support size. Since cuts strictly reduce or preserve $\mathcal{S}_t$, Hartley entropy is guaranteed to decrease monotonically. This makes $H_0$ the right quantity for our convergence analysis, while Shannon entropy will serve as an empirical diagnostic in experiments.

**Proposition 1 (Support Reduction).** Cuts are cumulative: $\mathcal{S}_{t+1} \subseteq \mathcal{S}_t$. Therefore
$$H_0(\mathcal{S}_{t+1}) \leq H_0(\mathcal{S}_t),$$
showing monotone entropy reduction.

**Remark (Shannon entropy as a diagnostic).** For the renormalized distribution, the Shannon entropy
$$H_1(\mathcal{S}_t) = - \sum_{P \in \mathcal{S}_t} \pi_\theta(P \mid C_t) \log \pi_\theta(P \mid C_t)$$
may increase if high-probability invalid modes are removed. Thus $H_1$ and more general Rényi entropies $H_\alpha$ are best viewed as empirical diagnostics, while $H_0$ provides the strict monotonicity guarantee.

**Witness sets and hitting sets.** Let $E$ be the set of edges. For any invalid path $P$, let $W(P) \subseteq E$ denote a *witness set* of edges such that every path containing $W(P)$ is invalid. Let $\mathcal{W} = \{W(P) : P \text{ invalid}\}$. A set $H \subseteq E$ is a *hitting set* if $H \cap W \neq \varnothing$ for all $W \in \mathcal{W}$. Let $H^\star$ be a minimum hitting set.

**Proposition 2 (Convergence via hitting sets).** Define $U_t$ as the number of uncovered witnesses at time $t$:
$$U_t = |\{W \in \mathcal{W} : W \cap E_t^{\text{forbid}} = \varnothing\}|,$$
where $E_t^{\text{forbid}}$ is the set of edges forbidden so far. If each cut $R_t$ intersects at least one uncovered witness, then $U_t$ decreases monotonically and satisfies $U_T = 0$ in at most $|H^\star|$ iterations. At that point either (i) a valid path is generated, or (ii) no invalid path remains feasible.

**Corollary (Cut strength).** If a cut $R_t$ hits $r_t \geq 1$ uncovered witnesses, then

$$U_{t+1} \leq U_t - r_t,$$

so the number of iterations is bounded by

$$T \leq \Big\lceil \frac{|H^\star|}{\min_t r_t} \Big\rceil.$$

Stronger cuts (larger $r_t$) accelerate convergence by eliminating multiple witnesses at once.

**Complexity and token costs** Each attempt incurs $L_0$ (base prompt) $+$ $L_{\text{gen}}$ (generation). Our reprompt header ``Avoid the following transitions in your next attempt:'' costs 19 tokens, and each weak cut u->v, costs 4 tokens (calibrated with the model tokenizer). Thus at round $t$: *naive* (print only new cuts) costs $19 + 4u_t$, while *cumulative cut-based* (reprint all cuts) costs $19 + 4U_t$ with $U_t = \sum_{k \leq t} u_k$. Combining with Prop. 2 ($T \leq |H^\star|/\bar{r}$) gives $\text{Tok}_{\text{naive}} = \mathcal{O}(T(L_0+L_{\text{gen}})+|H^\star|)$ and $\text{Tok}_{\text{cum}} = \mathcal{O}(T(L_0+L_{\text{gen}})+|H^\star|T) = \mathcal{O}(|H^\star|^2/\bar{r})$ in the worst case (cuts spread across rounds). For strong (node-level) cuts the per-round overhead scales with out-degree; constants and full derivations are in App. B.

**Failure modes and remedies.** If the model generates a path that is constraint-valid but does not reach $s_G$ (e.g., loops or subgoal divergence), then no witness in $\mathcal{W}$ is triggered and the process may stall. This can be mitigated by adding *progress witnesses*—constraints such as "no repeated states beyond $k$ visits" or "must reduce heuristic distance-to-go." Other remedies include escalating from weak to strong cuts when repeated failures occur, or hybridizing with heuristic search to bias exploration toward promising regions. These extensions are compatible with our framework and we leave a systematic study to future work.

# 4 METHODOLOGY

## 4.1 PROBLEM SETUP

We cast symbolic planning as path generation over a finite state machine (FSM). Formally, let $G = (V, E, s_0, s_G)$ denote a directed graph with state set $V$, edges $E \subseteq V \times V$, start $s_0$, and goal $s_G$. A valid plan is a path

$$P = [s_0, s_1, \ldots, s_G] \quad \text{such that } (s_i, s_{i+1}) \in E \ \forall i,$$

that connects the start to the goal without violating the transition structure. Transitions are treated as abstract actions, so planning reduces to generating a valid sequence of states.

We model the LLM as a stochastic policy $\pi_\theta$, conditioned on a context $\mathcal{C}$ that encodes the task:

$$P \sim \pi_\theta(\cdot \mid \mathcal{C}),$$

where $\mathcal{C}$ includes the FSM structure, start/goal nodes, and (optionally) feedback from prior attempts. This framing connects symbolic planning with sequence modeling: valid plans correspond to feasible trajectories under $\pi_\theta$.

## 4.2 CUT-BASED REPROMPTING

Single-shot prompting often fails on large graphs because the search space grows exponentially. To improve reliability, we introduce a *cut-based feedback loop* (Figure 1).

At each iteration, the LLM proposes a candidate path. A validator checks its validity: - If valid, the process halts successfully. - If invalid, the validator extracts offending edges and converts them into symbolic constraints (*cuts*), which are appended to the prompt.

The updated context conditions the next generation:

$$\mathcal{C}_{t+1} = \mathcal{C}_t \cup \textsc{MakeCuts}(\mathcal{F}_t), \quad P^{(t+1)} \sim \pi_\theta(\cdot \mid \mathcal{C}_{t+1}),$$

where $\mathcal{F}_t$ is the set of invalid edges at round $t$. This feedback loop progressively shrinks the feasible space and steers the model toward valid completions. The full procedure is summarized in Algorithm 1.

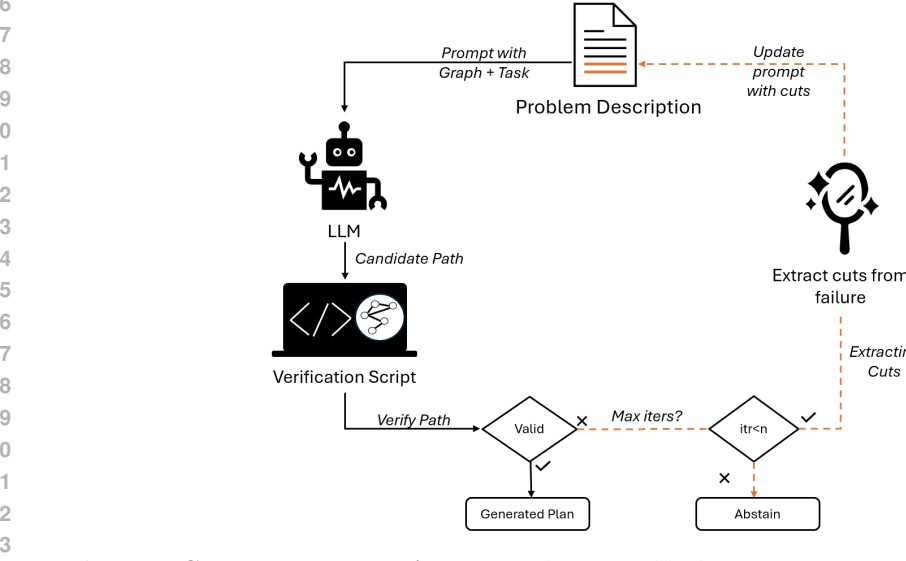

Figure 1: **Cut-based reprompting loop.** The LLM (black arrows) generates candidate paths. A validator checks validity; if invalid, symbolic cuts are extracted (orange arrows) and injected back into the prompt. This loop prevents repeated errors and progressively narrows the search space.

---

**Algorithm 1** Cut-Based Reprompting

---

**Input**: FSM $G = (V, E)$, start $s_0$, goal $s_G$, LLM $\pi_\theta$, budget $T$
**Output**: Valid path $P$ or failure
1: Initialize context $\mathcal{C} \leftarrow \text{PROMPT}(G, s_0, s_G)$
2: **for** $t = 1$ to $T$ **do**
3:     Sample $P \sim \pi_\theta(\cdot \mid \mathcal{C})$
4:     **if** $P$ valid **then**
5:         **return** $P$
6:     **end if**
7:     $\mathcal{C} \leftarrow \mathcal{C} \cup \text{MAKECUTS}(\text{INVALIDEDGES}(P, E))$
8: **end for**
9: **return** failure

---

### 4.3 WEAK VS. STRONG CUTS

A central design choice is how aggressively to prune the search space once an invalid path is detected. We study two regimes:

**Weak cuts.** A weak cut eliminates only the specific invalid transition observed in a failed path. For example, if the model proposes $2 \rightarrow 1$ but $(2, 1) \notin E$, the validator appends the instruction ``Do not choose $2 \rightarrow 1$''. Weak cuts are highly precise but conservative: the model may still attempt other invalid successors from the same node in later rounds.

**Strong cuts.** A strong cut generalizes feedback at the node level. Instead of forbidding one invalid edge at a time, the prompt enumerates the valid successors of a node and forbids all others. Formally, for state $s_i$:

"From node $s_i$, only transitions to ValidNext($s_i$) are allowed."

This prunes entire families of invalid paths in one step, at the cost of slightly longer prompts.

**Illustrative Example.** Consider the FSM $\{0 : [1, 2], 1 : [3], 2 : [3], 3 : []\}$. If the model outputs the invalid path $[0, 3]$: - A weak cut forbids only $(0, 3)$, preventing that exact error but leaving other invalid successors from node 0 untouched. - A strong cut enforces ``Node 0 only connects to [1,2]'', which removes all spurious transitions from node 0 at once.

In practice, weak cuts minimize feedback length, while strong cuts reduce the number of reprompts by eliminating multiple potential errors per iteration. This tradeoff underlies the complexity analysis in Section 3.

## 5 RESULTS

We evaluate cut-based reprompting on two case studies: (i) synthetic FSM traversal tasks, where we generate DAGs without isolated nodes to probe reasoning dynamics at scale, and (ii) MiniGrid navigation, a standard planning benchmark for embodied agents that tests generalization to spatial layouts and sequential dependencies.

### 5.1 ACCURACY ACROSS GRAPH SIZES

We measure planning accuracy as the percentage of valid paths successfully generated under different prompting and reprompting strategies. Results are shown in Figure 2.

**Impact of Graph Size.** Reprompting substantially improves accuracy compared to single-shot prompting, with cut-based strategies yielding the largest gains. On 100-node graphs, GPT-4o-mini rises from 26% (no reprompt) to 62% (Strong Cut) and 78% (CoT+Strong). Naive reprompting adds little benefit and often fails to converge on large graphs.

**Model Dependence.** Performance strongly depends on base model capacity. LLaMA-3-8B collapses beyond ∼30 nodes regardless of strategy, while GPT-4o-mini continues to benefit from reprompting. This shows that cuts amplify reasoning when the base model is capable, but cannot compensate for weak models.

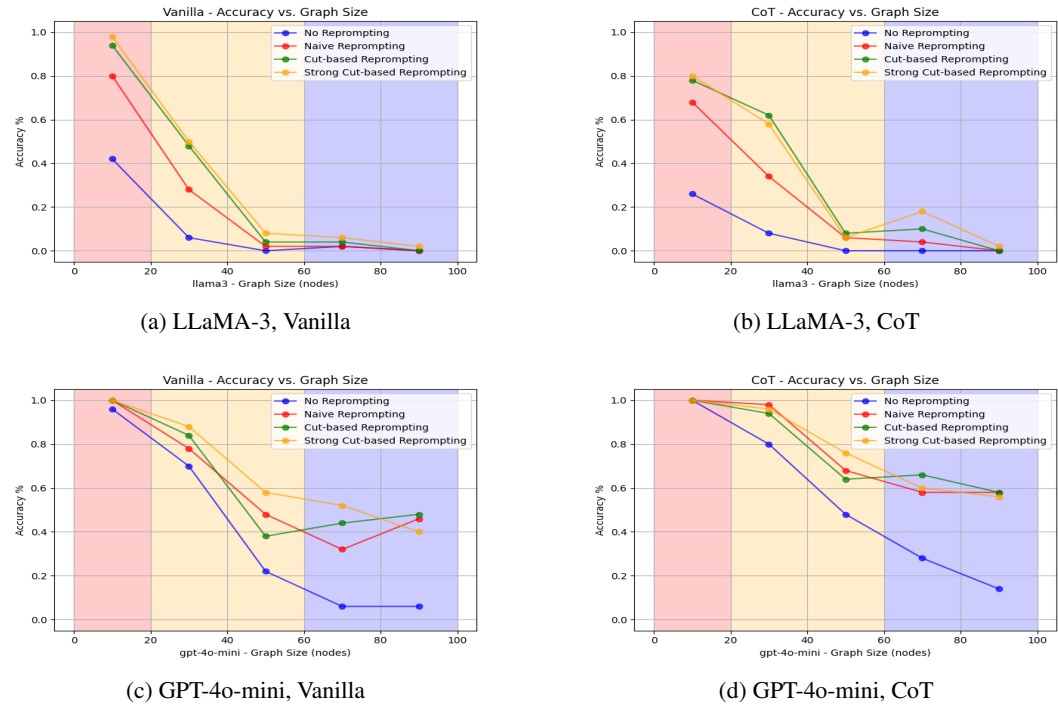

(a) LLaMA-3, Vanilla        (b) LLaMA-3, CoT

(c) GPT-4o-mini, Vanilla        (d) GPT-4o-mini, CoT

Figure 2: Accuracy vs. graph size across models and prompting strategies. Results are averaged over 10 runs per instance.

## 5.2 ENTROPY REDUCTION AND CONVERGENCE

To study convergence dynamics, we measure *path diversity*—the number of unique trajectories generated per reprompt step—as a proxy for entropy in the model's sampling distribution. Figures 3 plot diversity over 15 reprompt steps.

- **Naive reprompting** fails to reduce entropy: diversity remains high ($\sim$8–10 distinct paths) even after 15 steps.
- **Cut-based strategies** rapidly reduce entropy, collapsing to 2–3 consistent paths within 3–4 steps on 30-node FSMs.
- **Vanilla vs. CoT:** CoT produces smoother entropy decay and interacts synergistically with cuts; Vanilla sometimes plateaus at higher diversity.
- **Plateaus:** All methods stabilize after $\sim$8–10 iterations, reflecting persistent reasoning errors that reprompting alone cannot eliminate.

These dynamics confirm our theoretical claim: cuts act as an *entropy filter*, concentrating probability mass on feasible paths. CoT amplifies this effect, but convergence ultimately depends on base model capacity.

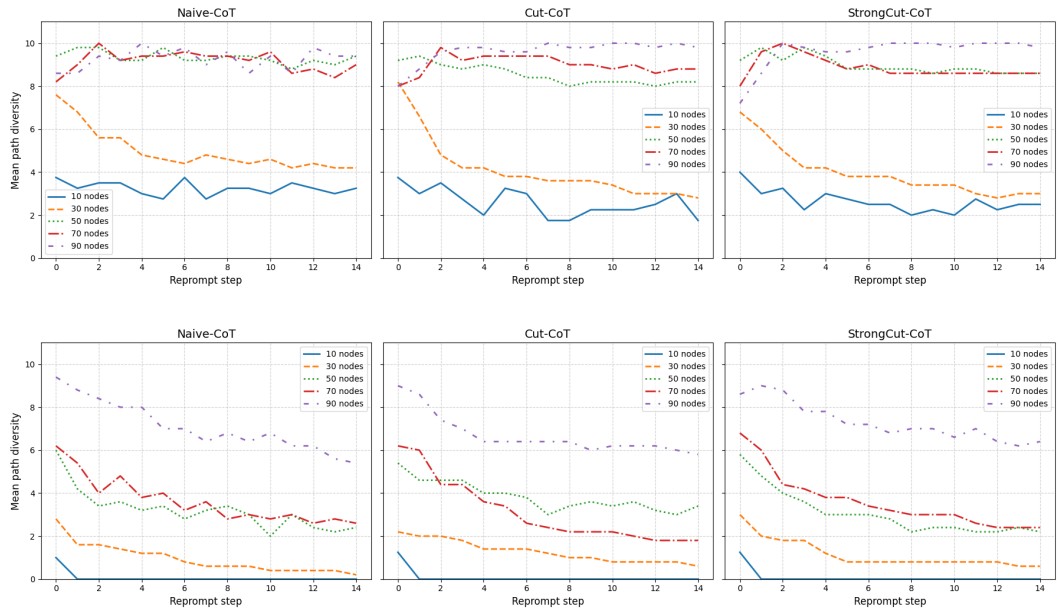

Figure 3: Entropy reduction across reprompting steps. Top: LLaMA-3. Bottom: GPT-4o-mini. Cut-based strategies consistently collapse path diversity, while naive reprompting leaves entropy high.

## 5.3 MINIGRID PATHFINDING

To test generalization beyond FSM traversal, we evaluate on MiniGrid navigation tasks of increasing grid size ($5 \times 5$, $6 \times 6$, $8 \times 8$, $16 \times 16$). Table 1 reports success rates for GPT-4o-mini and LLaMA-3-8B under Vanilla prompting with different reprompting strategies.

**Small grids** ($5 \times 5$). Both models benefit strongly from cuts. GPT-4o-mini achieves perfect accuracy (1.0) under any reprompting strategy, while LLaMA-3 rises only modestly ($0.3 \rightarrow 0.5$). This mirrors our FSM findings: entropy collapses effectively, but absolute success depends on base model strength.

**Medium grids** ($6 \times 6$). GPT-4o-mini continues to improve with stronger cuts ($0.4 \rightarrow 0.9$), whereas LLaMA-3 plateaus at or below 0.3 regardless of strategy. This highlights a synergy with base capacity: cuts can amplify competent reasoning but cannot rescue weaker policies.

**Larger grids** ($8 \times 8$ **and** $16 \times 16$). GPT-4o-mini maintains partial performance ($\approx 0.2$–$0.3$ for $8 \times 8$, $\approx 0.1$ for $16 \times 16$), while LLaMA-3 collapses to chance. Entropy still decreases monotonically under cuts, but the concentrated distribution is over incorrect paths. In other words, MiniGrid exposes a *capability ceiling*: cuts enforce symbolic consistency but cannot supply the missing spatial inductive biases required at larger scales.

| Model | Grid | No Reprompt | Naive | Cut-Based | Strong-Cut |
|-------|------|-------------|-------|-----------|------------|
| LLaMA-3-8B | $5 \times 5$ | 0.30 | 0.40 | 0.50 | 0.50 |
| | $6 \times 6$ | 0.20 | 0.30 | 0.30 | 0.30 |
| | $8 \times 8$ | 0.10 | 0.10 | 0.10 | 0.10 |
| | $16 \times 16$ | 0.00 | 0.00 | 0.00 | 0.00 |
| GPT-4o-mini | $5 \times 5$ | 0.80 | 1.00 | 1.00 | 1.00 |
| | $6 \times 6$ | 0.40 | 0.70 | 0.80 | 0.90 |
| | $8 \times 8$ | 0.20 | 0.20 | 0.30 | 0.30 |
| | $16 \times 16$ | 0.00 | 0.00 | 0.00 | 0.10 |

Table 1: MiniGrid success rates across grid sizes, reprompting strategies, and models. Reprompting substantially improves GPT-4o-mini on small/medium grids but fails to rescue LLaMA-3 beyond trivial cases.

## 5.4 Analysis and Insights

**Symbolic reliability.** Across both models, cuts consistently improve accuracy when the base policy has enough capacity, confirming our theoretical predictions. They act as an entropy filter that prunes incorrect transitions and concentrates mass on feasible regions.

**Feedback tradeoffs.** Weak cuts reduce redundancy gradually, while strong cuts eliminate entire families of errors in one iteration. GPT-4o-mini leverages these aggressively, but for LLaMA-3 the same cuts plateau early, reflecting insufficient base reasoning.

**Capability ceilings.** MiniGrid highlights a model-dependent ceiling. GPT-4o-mini solves up to $6 \times 6$ grids robustly but collapses beyond $8 \times 8$. LLaMA-3 collapses far earlier, at $5 \times 5$–$6 \times 6$. Entropy shrinks monotonically in all cases, but the residual distribution concentrates on wrong solutions when inductive biases are missing.

**Diagnostic value and hybridization.** We interpret this as diagnostic, not failure: symbolic cuts expose exactly where LLMs' reasoning suffices and where it breaks down. Beyond this ceiling, external grounding becomes essential. A natural next step is to hybridize with explicit planners (e.g., A*, value iteration) or learned transition models, where cuts enforce feasibility while the planner supplies missing spatial competence.

## 6 Discussion and Limitations

Cut-based reprompting provides a lightweight, model-agnostic reliability layer: by pruning invalid transitions, it reduces entropy monotonically and sharpens distributions toward valid solutions. This complements prompting strategies such as chain-of-thought while avoiding task-specific infrastructure.

**Capability ceilings.** Cuts enforce symbolic consistency but do not inject new semantic knowledge. As MiniGrid illustrates, when tasks depend on implicit world models (e.g., spatial layouts or key–door dependencies), accuracy collapses despite entropy reduction. In such cases, cuts converge to incorrect but consistent solutions—revealing the ceiling of the base model's reasoning ability. This diagnostic role is a feature: it makes clear when symbolic scaffolding suffices and when external grounding or inductive biases are necessary.

**Efficiency tradeoffs.**    Weak cuts conserve tokens but require more reprompts; strong cuts accelerate convergence at the cost of longer prompts. Adaptive strategies that balance these costs dynamically remain open, and our token-cost analysis (Appendix B) is a first step.

**Beyond symbolic costs.**    Symbolic pruning ensures feasibility but cannot capture qualitative distinctions between valid plans—for example, two routes of equal length where one is smooth and the other bumpy. Classical planning requires a hand-designed reward or cost function to encode such distinctions. LLMs, however, bring contextual priors and can often infer plausible preferences directly from descriptions. Cut-based reprompting complements this strength: cuts guarantee structural validity, while LLMs can supply reward-like judgments without explicit cost engineering.

**Baseline scope.**    We do not compare against grammar- or schema-constrained decoding, nor solver-in-the-loop hybrids. Our focus here was to isolate the general dynamics of reprompting via cuts in a model-agnostic setting. Constrained decoding is effective when low-level decoding hooks are exposed, but such methods are tied to local implementations and often unavailable in API-only models. In contrast, cut-based reprompting operates purely at the prompt–response level, requiring no architectural access. Solver hybrids offer stronger guarantees but change the problem by injecting explicit search; we defer such integrations to future work, focusing here on establishing reprompting as a stand-alone primitive.

**Real-world implementation.**    The framework hinges on the existence of a validator that can check a plan and generate cuts. For the tested FSM and MiniGrid problems, this is straightforward. However, for more complex, real-world planning problems this could present a challenge. That said many real-world applications are moving towards creating digital twins and simulation environments. These simulations are well-fit to play the role of a validator and have the same synergy with LLMs, where the forward problem of running a simulator is easy and the inverse-problem of coming up with the simulation inputs are non-trivial.

**Path forward.**    Scaling beyond controlled benchmarks will require hybridization with explicit planners (e.g., A*, value iteration) or learned transition models. In such settings, cuts ensure feasibility while planners supply optimality and semantic depth. Thus, the method is valuable both for improving reliability in symbolic domains and for probing the reasoning boundaries of current LLMs.

# 7    CONCLUSION

We framed large language models as stochastic policies for symbolic planning over finite state machines and introduced *cut-based reprompting*, a context-engineering technique that iteratively prunes invalid paths. Our theoretical analysis showed that symbolic cuts monotonically reduce entropy and guarantee bounded convergence under mild assumptions.

Empirically, cut-based reprompting improves planning accuracy and convergence across FSMs, with substantial gains for GPT-4o-mini, and entropy analysis confirmed that our method sharpens the distribution toward valid plans. Beyond improvements, our framework also served as a diagnostic lens: in MiniGrid, cuts reduced entropy yet revealed a capability ceiling, showing that spatial reasoning requires explicit world models rather than symbolic pruning alone.

Despite its simplicity, the method is robust, model-agnostic, and complementary to prompting strategies like chain-of-thought. Limitations include possible over-pruning and diminished benefits for weaker models. Future directions include adaptive cut strategies, hybridization with search and planning algorithms, and attention-based interpretability to better understand constrained decision-making.

By formalizing reprompting as entropy-based policy shaping and exposing capability boundaries, this work advances context engineering as a foundation for fault-tolerant reasoning and points toward hybrid symbolic–neural systems that combine the reliability of cuts with the semantic depth of planners.

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

## A  SUCCESS BY OPTIMAL PATH LENGTH ($L^*$)

**Setup.**  We group instances by their *optimal (shortest) path length* $L^*$ and report the fraction solved in each bucket (Success@$L^*$). An attempt counts as a success for bucket $L^*$ whenever the instance whose shortest path is $L^*$ is solved, regardless of the produced plan length. This evaluates robustness as a function of task difficulty rather than model behavior.

**Computation.**  For each strategy (None, Naive, Cut, Strong) and prompt type (Vanilla, CoT), we average binary success over all instances with the same $L^*$. Buckets with very few instances (e.g., $L^* \geq 12$) are sparse and thus noisier.

**Findings.**  Table 2 shows clear length sensitivity. (1) **GPT-4o-mini** remains strong on short plans ($L^* \leq 6$) across settings, and cut-based reprompting substantially slows the decay with length; *Strong* consistently dominates *Cut* at longer horizons. (2) **LLaMA-3-8B** benefits from cuts at small $L^*$ but collapses beyond $\sim 6$ steps, indicating a capacity ceiling that reprompting cannot overcome. (3) **CoT synergy**: CoT shifts the curve upward for both models, especially when combined with Strong cuts. (4) For $L^* \geq 10$, only GPT-4o-mini with cut-based strategies maintains non-trivial accuracy, highlighting current LLM limitations for long-horizon symbolic plans.

Table 2: Accuracy (%) by optimal path length $L^*$, model, prompt, and reprompting strategy. Deeper greens indicate higher success rates.

| Path Length | LLaMA 3:8B (Vanilla) | | | | LLaMA 3:8B (CoT) | | | | GPT-4o-mini (Vanilla) | | | | GPT-4o-mini (CoT) | | | |
|---|---|---|---|---|---|---|---|---|---|---|---|---|---|---|---|---|
| | None | Naive | Cut | Strong | None | Naive | Cut | Strong | None | Naive | Cut | Strong | None | Naive | Cut | Strong |
| 2 | 25.0 | 55.0 | 70.0 | 85.0 | 25.0 | 62.5 | 72.5 | 80.0 | 85.0 | 85.0 | 92.5 | 95.0 | 97.5 | 100.0 | 100.0 | 97.5 |
| 3 | 55.5 | 95.0 | 100.0 | 100.0 | 15.5 | 65.0 | 75.0 | 70.0 | 100.0 | 100.0 | 100.0 | 100.0 | 100.0 | 100.0 | 100.0 | 100.0 |
| 4 | 0.0 | 25.0 | 45.0 | 50.0 | 0.0 | 30.0 | 45.0 | 50.0 | 50.9 | 60.0 | 75.0 | 85.0 | 60.0 | 90.0 | 100.0 | 95.0 |
| 5 | 5.0 | 10.0 | 15.0 | 25.0 | 0.0 | 25.0 | 35.0 | 45.0 | 50.0 | 100.0 | 85.0 | 90.0 | 90.0 | 100.0 | 95.0 | 100.0 |
| 6 | 10.0 | 26.6 | 43.3 | 30.0 | 13.3 | 23.3 | 46.6 | 36.6 | 63.3 | 73.3 | 86.6 | 93.3 | 73.3 | 93.3 | 93.3 | 90.0 |
| 7 | 0.0 | 0.0 | 3.3 | 3.3 | 0.0 | 0.0 | 13.3 | 16.6 | 3.3 | 46.6 | 60.0 | 50.0 | 13.3 | 73.3 | 63.3 | 70.0 |
| 8 | 0.0 | 0.0 | 3.3 | 3.3 | 0.0 | 0.0 | 3.3 | 3.3 | 6.6 | 23.3 | 16.6 | 43.3 | 23.3 | 53.3 | 53.3 | 70.0 |
| 9 | 0.0 | 0.0 | 0.0 | 0.0 | 0.0 | 0.0 | 0.0 | 0.0 | 0.0 | 50.0 | 20.0 | 10.0 | 10.0 | 20.0 | 20.0 | 10.0 |
| 10 | 0.0 | 0.0 | 0.0 | 10.0 | 0.0 | 0.0 | 0.0 | 0.0 | 20.0 | 85.0 | 80.0 | 90.0 | 50.0 | 95.0 | 100.0 | 100.0 |
| 12 | 0.0 | 0.0 | 0.0 | 0.0 | 0.0 | 0.0 | 0.0 | 0.0 | 0.0 | 30.0 | 10.0 | 10.0 | 10.0 | 20.0 | 50.0 | 40.0 |
| 13 | 0.0 | 0.0 | 0.0 | 0.0 | 0.0 | 0.0 | 0.0 | 0.0 | 0.0 | 0.0 | 0.0 | 0.0 | 10.0 | 20.0 | 20.0 | 20.0 |
| 15 | 0.0 | 0.0 | 0.0 | 0.0 | 0.0 | 0.0 | 0.0 | 0.0 | 0.0 | 0.0 | 0.0 | 0.0 | 0.0 | 0.0 | 0.0 | 0.0 |

## B  TOKEN ACCOUNTING AND COMPLEXITY: FULL DERIVATIONS

We count tokens with the model's tokenizer. Each attempt (first try + reprompts) contains: (i) the base prompt $L_0$ (task, graph, etc.), (ii) the reprompt header, (iii) the cut text, and (iv) an average generation of $L_{\text{gen}}$ tokens.

**Fixed strings and per-cut cost.**  Our reprompt header

```
``Avoid the following transitions in your next attempt:''
```

consumes a fixed $c_0 = 19$ tokens. Each weak cut is printed as `u->v,` at a calibrated cost $c_w = 4$ tokens. We denote by $u_t$ the number of *new* weak cuts added at round $t$, and by

$$U_t = \sum_{k=1}^{t} u_k$$

the *cumulative* number of distinct weak cuts by the start of round $t+1$. Let $T$ be the number of attempts (first attempt counted as $t=1$).

### B.1  WEAK CUTS: NAIVE VS. CUMULATIVE PRINTING

**Naive reprompting (new cuts only).**  At round $t$, the reprompt overhead is

$$L_{\text{weak}}^{\text{naive}}(t) = c_0 + c_w \, u_t = 19 + 4 \, u_t.$$

The total token budget over $T$ attempts is

$$\text{Tok}_{\text{naive}} = \sum_{t=1}^{T} \left(L_0 + L_{\text{weak}}^{\text{naive}}(t)\right) + T\,L_{\text{gen}} = T(L_0 + L_{\text{gen}}) + \sum_{t=1}^{T} \left(19 + 4\,u_t\right). \tag{1}$$

**Cumulative cut-based (all cuts discovered so far).** At round $t$, the reprompt overhead is

$$L_{\text{weak}}^{\text{cum}}(t) = c_0 + c_w\,U_t = 19 + 4\,U_t.$$

The total is

$$\text{Tok}_{\text{cum}} = \sum_{t=1}^{T} \left(L_0 + L_{\text{weak}}^{\text{cum}}(t)\right) + T\,L_{\text{gen}} = T(L_0 + L_{\text{gen}} + 19) + 4\sum_{t=1}^{T} U_t. \tag{2}$$

Reordering the double sum gives the useful identity

$$\sum_{t=1}^{T} U_t = \sum_{t=1}^{T}\sum_{k=1}^{t} u_k = \sum_{k=1}^{T} u_k\,(T - k + 1). \tag{3}$$

Hence

$$\text{Tok}_{\text{cum}} = T(L_0 + L_{\text{gen}} + 19) + 4\sum_{k=1}^{T} u_k\,(T - k + 1). \tag{4}$$

Subtracting equation 1 from equation 4 yields the overhead relative to naive:

$$\text{Tok}_{\text{cum}} - \text{Tok}_{\text{naive}} = 4\sum_{k=1}^{T} u_k\,(T - k) \geq 0. \tag{5}$$

If cuts are added roughly uniformly across rounds, each cut is reprinted on average $\approx (T+1)/2$ times.

## B.2 Hitting-set convergence $\Rightarrow$ token bounds

Let $H^\star$ be a minimum hitting set over witnesses (§3). If each cut hits at least $\bar{r} \geq 1$ previously uncovered witnesses, Prop. 2 gives the attempt bound

$$T \leq \left\lceil \frac{|H^\star|}{\bar{r}} \right\rceil.$$

Moreover, because each weak cut covers at least one witness,

$$\sum_{k=1}^{T} u_k \geq |H^\star|.$$

**Naive (new cuts only).** From equation 1,

$$\text{Tok}_{\text{naive}} \leq T(L_0 + L_{\text{gen}} + 19) + 4\,|H^\star| = \mathcal{O}\Big(T(L_0 + L_{\text{gen}}) + |H^\star|\Big).$$

With $T \leq |H^\star|/\bar{r}$, this is linear in $|H^\star|$.

**Cumulative (all cuts discovered so far).** Using equation 4 and the fact that $(T - k + 1) \leq T$,

$$\sum_{k=1}^{T} u_k\,(T - k + 1) \leq T\sum_{k=1}^{T} u_k \leq T\,|H^\star|.$$

Thus,

$$\text{Tok}_{\text{cum}} \leq T(L_0 + L_{\text{gen}} + 19) + 4\,T\,|H^\star| = \mathcal{O}\Big(T(L_0 + L_{\text{gen}}) + |H^\star|\,T\Big) = \mathcal{O}\Big(|H^\star|^2/\bar{r}\Big)$$

in the worst case (cuts spread across rounds). If most cuts arrive late (large $k$), the overhead factor in equation 5 is smaller; if most arrive early, it is larger (upper bounded by $T$ per cut).

### B.3 STRONG (NODE-LEVEL) CUTS

A strong cut enumerates valid successors for a node $i$:

$$\text{From node } i, \text{ only transitions to } [v_1, \ldots, v_{d_i}].$$

Let $d_i$ be the out-degree of $i$. The per-node cost can be parameterized as

$$c_s(i) \approx c_s^{(0)} + c_s^{(1)} d_i,$$

where $c_s^{(0)}$ is the fixed phrase cost and $c_s^{(1)}$ the per-successor cost (identifier + separator). If $\mathcal{N}_t$ is the set of nodes newly strong-cut at round $t$, the added text at $t$ is

$$L_{\text{strong}}(t) = c_0 + \sum_{i \in \mathcal{N}_t} \left( c_s^{(0)} + c_s^{(1)} d_i \right).$$

In graphs with maximum out-degree $\Delta$, $c_s(i) \leq c_s^{(0)} + c_s^{(1)}\Delta$, so

$$L_{\text{strong}}(t) \leq c_0 + |\mathcal{N}_t| \left( c_s^{(0)} + c_s^{(1)}\Delta \right).$$

If each strong cut covers at least $g \geq 2$ witnesses (node-level coverage), then $T \leq |H^\star|/g$, and the total token cost satisfies

$$\text{Tok}_{\text{strong}} \leq T(L_0 + L_{\text{gen}} + 19) + \sum_t \sum_{i \in \mathcal{N}_t} \left( c_s^{(0)} + c_s^{(1)} d_i \right),$$

exhibiting the trade-off: larger per-round cost offset by fewer rounds $T$ (improved coverage $g$). In mixed policies (weak + strong), if $U = \sum_t u_t$ and $S = \sum_t |\mathcal{N}_t|$, then $gS + U \geq |H^\star|$ and $T \leq U + S$, making the cost/coverage trade-off explicit.

### B.4 CONSTANTS AND CALIBRATION

We calibrated $c_0 = 19$ (header) and $c_w = 4$ (weak cut) with the model tokenizer using the exact strings above and verified them in logs. For strong cuts, we estimate $(c_s^{(0)}, c_s^{(1)})$ by regressing per-iteration prompt growth on the number of newly strong-cut nodes and their out-degrees; the fitted values can be treated as constants for a fixed phrasing and tokenizer.

## C  REPRODUCIBILITY STATEMENT

Our experiments involve stochastic language model outputs (sampling with temperature = 1), so individual trajectories may vary across runs. To ensure reproducibility, we (1) release all code, FSM benchmarks, and MiniGrid configurations in the supplementary material, to be made public upon acceptance; (2) report results averaged over 10 instances; and (3) document all prompt templates and validation rules in the code. While exact outputs cannot be reproduced deterministically, the aggregated results and code allow independent researchers to replicate our findings within statistical variance.

## D  LLM USAGE DISCLOSURE

Large language models (LLMs) were used during the preparation of this paper in the following ways:

Writing assistance: LLMs were used to help with editing, phrasing suggestions, and polishing for clarity and concision.

Reviewer simulation: LLMs were used to generate mock peer reviews to anticipate possible reviewer concerns.

No role in experiments or analysis: All experiments, data generation, theoretical derivations, and result analysis were conducted entirely by the authors.

The final content, claims, and conclusions are the responsibility of the authors.

