# OpenReview forum: "Symbolic Planning Using LLM Agents: A Cut-Based Reprompting Approach"
_ICLR.cc/2026/Conference — Submitted to ICLR 2026_

### Official Review · Reviewer_HF9f · 2025-10-27

**Soundness:** 2
**Presentation:** 2
**Contribution:** 2
**Rating:** 2
**Confidence:** 4

**Summary:**

The paper presents a methodology to use LLMs in planning and reasoning tasks while offering correctness guarantees of the result. The core idea is wrapping the LLM in a verify & refine loop, where verify checks whether the current LLM response solves the given problem according to a symbolic FSM representation of the problem, and refine augments the prompt accordingly in case the check failed. A preliminary empirical evaluation demonstrates the approach on synthetic FSM benchmarks and small Minigrid instances.

**Strengths:**

LLMs are receiving increasing attention as solvers of planning and reasoning tasks. By their very design, however, LLMs by themselves cannot provide any kind of guarantee for the solutions it finds. The present paper proposes a conceptually simple, yet in principle, powerful idea to tackle this limitation. The paper write-up is overall very clean, and to a large extent easy to follow. Reproducibility standard should be satisfied.

**Weaknesses:**

While the principle idea is promising, in the current form, the paper suffers from major problems in terms of (1) novelty, (2) motivation and clarity, (3) soundness, and (4) breadth of experiments; resolving which requires in my opinion a major paper revision.

1. Novelty: The idea of using symbolic verification methods to validate LLM output, more specifically plans computed by the LLM, and automatically refine prompts as necessary is not novel and has already been explored before [1, 2]. For example [1], in a nutshell, considers symbolic models of the planning problem, which is not unsimilar to the assumption of an FSM representation in this paper. This allows syntactic checking of the validity of the computed plans, and upon finding an error, allows the automatic generation of prompt refinements to avoid making similar errors again. In fact, by using a formal but higher level representation language than FSM, makes their approach much more scalable and much easier to use compared to the proposed FSM-based method.

2. Motivation and clarity. The authors' text motivating the introduced approach contains obvious logical contradictions. On the one hand, they argue that due to their requirement of a symbolic model causes major practical limitations of pure symbolic approaches. On the other hand, however, their approach does itself come with the very same requirement; even worse, it comes with the specific requirement of a finite-state machine. Hence, it is not really clear in how the proposed approach should be any better than pure symbolic methods. In general, it is also not really clear what form the FSM needs to be in exactly. From the description, I would deduct that the FSM needs to be available in a concrete graph form (enumerating all vertices and edges). For all non-trivial planning/reasoning problems, this is clearly not feasible. Also, this raises the question of why not searching for a solution in the FSM directly? In fact, one of the proposed methods -- strong cuts -- for refining the LLM prompts seems to do exactly that, namely in order to find the set of all edges to avoid in a particular vertex. So if it is possible to compute solutions from the FSM directly, why then using the LLM at all?

3. Soundness. Due to the general lack of information on the assumptions in general and the method's intrinsic in particular, the discussion of the theoretical properties of the proposed method is hard to follow and, critically, it is not possible to validate the claims. For instance, it is not clear why the claimed monotonicity and termination bound properties indeed hold. Plain LLMs do not provide any guarantees on their output, and in particular, one can a priori not establish any relation between the returns for two, even almost identical, but still different prompts. Hence, if there is no other prevention method in place, it is not clear why the LLM would not return the same invalid solution at some later iteration; which, if the case, invalidates any monotonicity or termination property. Secondly, the discussion of the costs of the prompt fixes is not clear either, given that there is no particular information about the prompting specifics; and the cost function assessment heavily makes use of a particular tokenization, which is however never introduced. Given the lack of details, it is not possible (for me) to verify the correctness/soundness of the claims. Finally, the authors claim that their approach can be seen as a method to reduce the entropy of the solution space, i.e., move probability mass onto specific solutions, implicitly present in the LLM. This is clearly not true (the authors give an example of why this is not the case themselves); and the authors cheat by choosing some random measure, which they call entropy, but really has nothing to do with entropy in the actual sense.

4. Experiments. The experimental validation is far too narrow. In the tested benchmarks are so simplistic that it is not even clear at all why an LLM should be used here in the first place. The experiments also lack any kind of scalability study, information on the size of the FSMs, etc., which makes it impossible to judge the practicality or at least the potential of the method. Moreover, it is not clear why related approaches have not been compared to, e.g., the works cited below. As it stands, the experiments need to be significantly extended in order to demonstrate the utility of the proposed methodology.

References

[1] Arora, Daman, and Subbarao Kambhampati. "Learning and leveraging verifiers to improve planning capabilities of pre-trained language models." arXiv preprint arXiv:2305.17077 (2023)

[2] Daniel Cao, Michael Katz, Harsha Kokel, Kavitha Srinivas, Shirin Sohrabi: Automating Thought of Search: A Journey Towards Soundness and Completeness. CoRR abs/2408.11326 (2024)

**Questions:**

1. Can you defend your work against my criticisms in point 2?

2. Can you provide me some more details regarding your overall method and use of the LLM to generate the solution plans; and why this guarantees that your monotonicity and termination analysis is valid?

3. How are FSM handled exactly, how is the problem formulated to the LLM and how are cuts then represented in the prompt? How big are the FSM considered in your experiments, and how did you represent the Minigrid benchmarks in this FSM framework?

---

### Official Review · Reviewer_KoDm · 2025-10-28

**Soundness:** 2
**Presentation:** 3
**Contribution:** 2
**Rating:** 2
**Confidence:** 3

**Summary:**

The main contribution of this paper is an iterative prompting approach for symbolic planning wherein an LLM is coupled with a symbolic validator: if the validator finds a constraint violation (e.g., the plan traverses an edge that does not exist), then the next prompt is appended with these violations so the LLM will (hopefully) generate a better (valid) plan. The paper provides some theoretical support under several (reasonable to me) assumptions, showing that this cut-based reprompting approach monotonically reduces policy entropy. Experiments over synthetic finite-state-machine traversal tasks and MiniGrid navigation compare two versions of the proposed approach vs. no reprompting and a naive reprompting.

**Strengths:**

+ The main idea is conceptually quite straightforward and makes a lot of sense as a general-purpose approach to improving symbolic planners.

+ I like that there is effort to provide theoretical support for cut-based reprompting. The assumptions made all seem reasonable to me (I guess we could quibble if cuts will always be faithfully incorporated into the context), and there is demonstration that the cut-based approach monotonically reduces policy entropy and guarantees finite-step convergence.

**Weaknesses:**

- My biggest concern is the experimental study -- both what is presented and how it is characterized.

 -- First, the experiments focus on (to my eyes) quite small tasks: graphs of size less than 100 and MiniGrid traversals on graphs of up to 16x16 size. Further both these scenarios are not clearly described. What is the nature of the graphs in terms of complexity? How were they generated? What are the MiniGrid scenarios specifically? All in all, it is difficult to appreciate the impact of the proposed methods without a larger-scale study.

 -- Second, the presentation of the results is difficult to parse (at least for me). For example, Page 6 discusses 100-node graphs but all the figures only go up to 90 nodes. Strange. And the discussion says GPT-4o-mini rises from 26% (no reprompt) to 62% (strong cut) and 78% (CoT+Strong). Looking at the figures, I cannot find points that correspond to any of these numbers. Regardless, if we focus on Figures 2c and 2d, the results are a bit ambiguous: reprompting does improve over no reprompting, but there are many cases where the naive reprompting works quite well.

 -- Continuing to Section 5.2, it is asserted that naive remprompting has around 8-10 distinct paths, but cut-based strategies collapse to 2-3 paths with 3-4 steps on 30 node graphs. But looking at Figure 3, yes, naive reprompting has 8-10 paths for 50, 70, 90 node graphs, but around 4 paths for 30 node graphs. I is not clear why the results are presented in such a way.


I don't see a proper definition of "naive reprompting" (maybe I missed it). From the appendix I guess it is just reprompting with the current cuts? And not remembering the previous cuts? It would be good to highlight this more clearly in the body of the paper.

**Questions:**

Could you clarify the presentation of the results?

---

### Official Review · Reviewer_2HWB · 2025-10-30

**Soundness:** 3
**Presentation:** 1
**Contribution:** 1
**Rating:** 0
**Confidence:** 4

**Summary:**

This paper proposes cut-based reprompting, a method that integrates a symbolic validator into an LLM loop for tasks  finite state machine (FSM) traversal tasks. The LLM proposes a path in the FSM, the validator checks its validity, and when failures occur, symbolic cuts (constraints forbidding invalid transitions) are added to the prompt. The authors formalize this as an entropy-reducing process. Empirical results on FSM traversal and MiniGrid tasks show that the technique improves success and convergence speed compared to vanilla reprompting, especially for more larger models (GPT-4o-mini).

**Strengths:**

The proposed method improves performance.

**Weaknesses:**

My main concern is that the conclusion from this paper is already well-known and has been exploited by the planning community (see [1] as the first example, but with problems which are much harder than yours). I like the framing of "cut-prompting" but, at the end of the day, the idea is just the same as what other people getting the feedback from VAL.

I appreciate the authors trying to formalize the impact of their approach theoretically. But the theoretical results are not really useful, nor do they fit in the context of LLMs. Also, there's a lot of details missing and the notation does not fit properly. Overall, all definitions and result seem to incomplete to me. The idea of convergence via hitting sets basically prunes parts of the solution space. But this is not useful in the context of inference with LLMs. What would be actually interesting would be to show how the distribution probability of the tokens converge, but this is not what is there. Additionally, the result discussed in the paper seems trivial for any combinatorial problem: if you discard invalid solutions from your candidate solution space you eventually end up only with valid solutions.

Another concern that I have is that the paper is underspecified. For example, I had to look inside the source code to find out how the prompts look like. Again, the theoretical results also deserve more attention and must be expanded in more detail.

The main premise is also problematic. To start, casting symbolic planning as an FSM doesn't make sense. (The only way you can correctly represent symbolic planning problems as FSM is if you factor them as sets of FSMs and then compute an accepting word for all of them, i.e., compute a word accepted by the intersection of the FSMs. This is well-studied in the symbolic planning literature as well. But this is much harder than what you propose.) So while the paper is phrased in terms of planning, it should really be "pathfinding over an explicit graph". The bit about defining the prompt as a a distribution over candidate plans, and decode corresponds to sampling from it is interesting but I am not sure this actually means anything in practice. The distribution of the output for a fixed prompt is not a direct mapping to candidate plans---it is a direct mapping to tokens sequences, and some of them might be plans.

The paper also contains some parts that seem just wrong or that made no sense to me, for example:
- line 89: you say that cut-based reprompting is solver-free, lightweight, and model-agnostic while other approaches are not. However, the other approaches use the same ideas as you --- LLM + symbolic validator.
- paragraph of line 460: what does "semantic depth" means in the context of reasoning and A*-search?

Last, but not least, the paper contains some sloppy statements and presentation that just cause confusion. For example, the abstract talks about tasks where validating the solution is cheap but computing one is hard, as if this is a significant restriction. But this is *literally* the entire NP class. I also found the related work confusing, as it discussed lots of things which are not related work (e.g., "Our Contribution"). Another issue is that the experimental setup is never properly specified, and the text sometimes reports results which cannot be observed in the plots (e.g., the case for 100 nodes).

[1] Karthik Valmeekam, Matthew Marquez, Sarath Sreedharan, Subbarao Kambhampati:
On the Planning Abilities of Large Language Models - A Critical Investigation. NeurIPS 2023

**Questions:**

I have no specific questions, but please correct any wrong statement from my Weaknesses section.

---

### Official Review · Reviewer_WxA5 · 2025-10-31

**Soundness:** 3
**Presentation:** 2
**Contribution:** 2
**Rating:** 4
**Confidence:** 3

**Summary:**

This paper proposes cut-based reprompting, a method to improve the reliability of LLMs in symbolic planning tasks. The method combines an LLM with a symbolic validator. When the LLM proposes an invalid plan, the validator generates structured constraints, cuts, that forbid the observed failure pattern. The authors provide a theoretical framework framing LLMs as stochastic policies and show that their method monotonically reduces the policy's Hartley entropy. They evaluate the approach on synthetic FSM traversal and MiniGrid environments using GPT-4o-mini and LLaMA-3-8B, showing higher success rates and faster convergence compared to naive or vanilla reprompting.

**Strengths:**

* The paper provides a formal analysis in addition to an empirical demonstration. Framing the problem in terms of entropy reduction and providing convergence guarantees strengthens the contribution.
* The concept of using symbolic cuts for iterative reprompting is simple but powerful.
* Experiments demonstrate clear and interpretable performance. The evaluation and analysis are comprehensive.
* The paper thoughtfully discusses the method's limitations

**Weaknesses:**

* Although the authors provide detailed theoretical analysis, cut-based reprompting is inherently a simple technique that reprompt with external execution feedback, which is not algorithmically novel.
* Though admitted, the assumption that the validator exists still weakens the contribution.
* The paper’s clarity could benefit from more intuitive examples or visualizations.
* The baselines mainly include direct prompting or naïve reprompting. Although the authors claim they focus to reprompting methods, more baselines such as self-reflective/ environment feedback method could also be used as reprompting methods.

**Questions:**

* Could the authors clarify whether the main contribution of this work lies in introducing the cut-based reprompting algorithm itself, or primarily in formalizing and theoretically analyzing reprompting methods? From the current presentation, the algorithmic novelty appears modest, and the comparison with other reprompting methods seem not comprehensive.
* On 100-node graphs, the performance gain of cut-based reprompting is not obvious compared to naive reprompting. Sometimes even naïve reprompting has better performance. Could the authors provide some analysis?

---

### Meta-Review · Area_Chair_2XcK · 2026-01-09

**Summary:**

All reviewers suggest rejection and the authors didn't provide any rebuttal.

**Reviewer Concerns:**

All reviewers' concerns remain unaddressed.

**Reviewer Scores:**

All reviewers give negative scores (0,2,2,4).

---

### Decision · Program_Chairs · 2026-01-26

Reject